# A Simple, Low-Cost Micro-Coating Method for Accuracy Improvement and Its Application in Pressure Sensors

**DOI:** 10.3390/s19204601

**Published:** 2019-10-22

**Authors:** Jia-Lin Yao, Yu-Xuan Chen, Wen-Jiang Qiang, Xi-Zi Wang, Hao Wei, Bo-Hang Gao, Xing Yang

**Affiliations:** 1School of Materials Science and Engineering, University of Science and Technology Beijing, Beijing 100083, China; colin-yao@foxmail.com (J.-L.Y.); ustbcyx@163.com (Y.-X.C.); wjqiang@mater.ustb.edu.cn (W.-J.Q.); wxz41604206@163.com (X.-Z.W.); 18800167020@163.com (H.W.); 18810580357@163.com (B.-H.G.); 2The State Key Laboratory of Precision Measurement Technology and Instruments, Department of Precision Instrument, Tsinghua University, Beijing 100084, China

**Keywords:** micro-coatings, accuracy improvement, pressure sensors, post-processing methods, protective barrier

## Abstract

The demand for high-accuracy pressure sensors has increased with the advancement of technology in a wide variety of applications. However, it is generally difficult and expensive to improve the accuracy of the pressure sensor because it usually depends on the sensing principle and the internal physical structure of the pressure sensor, varying with its material and production process. Thus, a simple, low-cost, and generally applied post-processing method is proposed to improve the accuracy of pressure sensors. In this method, a micro-coating is cladded on the surface of the sensor, which effectively isolates the adverse effect of the external environment, similar to applying a “micro-protective clothing” on the pressure sensor. Experiments on seven pressure sensors are conducted, in which the micron-thin parylene polymer is utilized as the surface-deposited coating layer to demonstrate the improvement of accuracy. Results show that the accuracy was improved, with an average increase of approximately 62.54% than before cladding, while the sensitivity was almost unchanged. The principle of improving the accuracy of this method was also analyzed. The proposed simple, efficient, and low-cost method of cladding micro-coating for enhancing the accuracy of sensors can be widely applied in various fields of industrial automatic control.

## 1. Introduction

Pressure sensors are among the most commonly used sensors in the industry, including microfluidic, medicine, aerospace, and other industries [1,2,3,4,5]. With the development of technology, high-accuracy pressure sensors are highly desired in a variety of applications. In the medical field, various types of implantable pressure sensors (such as blood and intraocular pressure sensors) have requirements which need to be very highly accurate [6,7,8,9]. The accuracy of implantable blood pressure sensors is required to be about ±1 mmHg, and the accuracy of implantable intraocular sensors should be even better than ±1 mmHg [10]. In the field of aerospace, the altimeter of an aircraft determines the flight altitude of the aircraft through small changes in air pressure, and the accuracy of some unmanned aerial vehicle altimeter needs to reach 0.01 mmHg [11,12,13].

However, improving the accuracy index of pressure sensors is difficult and expensive. Existing technologies generally consider the sensing principle, the internal physical structure of a pressure sensor, material, production process [14,15,16,17,18] (e.g., manufacturing errors and inhomogeneity of the pressure sensor material), which will increase the difficulty of improving the accuracy index. Starting from the principle and structure, the General Electric Company (GE) utilizes groove etching silicon resonant pressure sensor technology to achieve high-accuracy pressure monitoring (~0.1 mmHg), but the device production process is complex, resulting in a longer manufacturing cycle, which also increases the price to thousands of US dollars [19]. Therefore, the methods of improving the accuracy of the device from the internal physical structure of the pressure sensor are generally complicated and costly.

In addition to the internal physical structure of a sensor, the corrosion and interference of the application environment of sensors significantly affect the accuracy index [20,21,22]. In terms of long-term influence, with the change of the external environment over time, the materials and the properties of the pressure sensor will also change, resulting in a decrease in accuracy of the pressure sensor due to corrosion induced by moisture, oxygen in the air, and other media. As for short-term effects, the interference on the pressure measurement caused by the temperature and humidity around the device is also an external factor affecting the accuracy. However, related techniques on improving the accuracy of the pressure sensor by isolating the influence of the external environment have not been reported.

Therefore, a simple and low-cost method is proposed to improve the accuracy index of pressure sensors. Based on the previous studies, we consider the impact of reducing the factors related to the external environment on the pressure sensor. This post-processing method, through cladding coating in the external surface of the pressure sensor, can reduce the interference and influence of the external environment, similar to applying a “micro-protective clothing” on the pressure sensor.

The micro-coating, which improves accuracy, needs to meet the following three basic requirements:

(1)Good blocking and isolation capabilities from influential media in the external environment.(2)Deposition of densified pinhole-free coating to the surface of the pressure sensor to avoid errors caused by loosening between the surface of the device and the surface of the coating.(3)The lightweight and soft structures are desirable, because a heavy coating will directly affect the flexibility of the movable structure of the pressure sensor. Thus, a lightweight coating material with small Young’s modulus is necessary. Besides, the lightweight coating can fill and make up for defects, such as surface clearance.

With the development of microfabrication technology, we were able to prepare micro-coatings to meet the above three requirements so as to improve the accuracy of the sensors. Therefore, parylene-C, a polymer material with good barrier protection ability, flexibility, and biocompatibility was selected. Parylene-C coating has excellent barrier protection capability in acidic, alkaline, and organic solvents, or in other harsh environments, such as water vapor and salt mist. Parylene-C coating also has low thermal conductivity [10,23], so it can reduce the interference of ambient temperature change, which affects the linear output of the pressure sensor to some extent. Given that the gas phase deposition is performed under vacuum conditions, the active small molecules of parylene-C can grow in cracks, inner surfaces, and edges of the substrate material, so the pinholes can be eliminated when the coating is micron-thick [24,25].

The principle of the proposed method, which improves the accuracy of the pressure sensor by micro-coating, is demonstrated in Figure 1. As is shown in Figure 1a,b, the pressure sensor was coated with a micro-coating (similar to micron protective clothing), which can reduce the intrusion of the medium and provide a better-confined environment, and in turn, improve the indicators related to accuracy. We applied parylene-C in our experiment, and the chemical structure (molecular diagram) is shown in Figure 1c. A micron-thin coating cladding of the pressure sensor’s post-processing method improved the accuracy of the device and other performance indicators. The coating deposited by this method also had good blocking, isolation ability, and light structure.

## 2. Materials and Methods

### 2.1. The Specific Steps of the Post-Processing Method

The SMI SM5240 pressure sensors with a pressure range of 0~15 psi absolute were utilized in the experiment as an example. As shown in Figure 1d, the post-processing method includes three steps. Step 1 is cleaning the pressure sensors with isopropyl alcohol (IPA, ≥99.7%, Sigma-Aldrich Corporation, St. Louis, MI, USA). Step 2 is leading wires to the pressure sensor for examination. Finally, Step 3 is cladding the micro-coating on the surface of the sensor. A parylene special coating system (Specialty Coating Systems, Inc., Indianapolis, IN, USA) PDS2010 was utilized to form the micro-coat protective layer on the pressure sensor. 2.16 g of solid raw material C-type parylene dimer (Specialty Coating Systems, Inc., Indianapolis, IN, USA) was placed into a PDS2010 evaporation chamber at 175 °C for heating and sublimation under 0.1 torr. The gaseous dimer obtained through sublimation entered the cracking chamber of PDS2010 and was then cleaved into the p-xylene monomer at 690 °C and 0.5 torr. The gaseous monomer of p-xylene entered the PDS2010 vacuum deposition chamber at room temperature and 0.1 torr, and then deposited and polymerized on the surface of the device to a micron-thin parylene-C protective layer.

### 2.2. Characterization of the Micro-Coating Film

For measurement of the coating thickness, the United States Dektak XT step profiler (where its height repeatability is <5 Å; the vertical resolution is 1 Å) was utilized; on top of three pieces of silicon wafers, which were clad with the micro-coating together with pressure sensors in the experiment, five points were randomly selected to test under the mentioned step profile to get the thickness data of the coating [25,26]. The average coating thickness was 0.95 ± 0.1 μm. In addition, to characterize the morphology of the micro-coating, the pressure sensor coated with a micro-coating was cut with tweezers and knives. The coating was very thin (~1 μm), as shown in Figure 2a,b. The coating morphology on the silicon substrate was further observed, as shown in Figure 2c,d, in which the coating surface was dense without voids and cracks.

### 2.3. The Method to Test the Pressure Sensor

At room temperature (25 ± 1 °C), the pressure sensor to be measured was placed in a sealed pressurized compartment. A preload of not less than three times to the sensor was applied, and then uniform six points were selected within the full range, including the sensor measurement’s upper and lower limits, to measure the sensor output corresponding to the input pressure point. Our pressure measurement ranged from 0 to 75 mmHg of the gauge pressure; the pressure division was 15 mmHg, the sensor output corresponding to the input pressure point was recorded, and the up-and-down calibration cycle was repeated for at least three times.

The test data was processed with reference to the standard GBT 15478-2015. The specific calculation method of accuracy included the following three steps:

(1)Determine the characteristic equation of the linear sensor. By using the least-squares calculation and the characteristic equation of the linear sensor $y_{i}=k\times x_{{}_{i}}+a$
was obtained. Then the intercept a was calculated by utilizing the intercept function in MS Excel software, the slope was calculated by using the slope function k, where $x_{i}$ is the pressure input value of the measuring point of i, and $y_{i}$ is the signal output value of the measuring point i of the sensor.(2)Determine the output value of full-scale $Y_{FS}$: $Y_{FS}=k\times (x_{H}-x_{L})$ was calculated by using the least-squares method, where $Y_{FS}$ is the least-squares method to calculate the maximum range output, $x_{H}$ measures the upper limit pressure value, and $x_{L}$ measures the lower limit pressure value.(3)To calculate the accuracy value of the sensor, $\xi_{L}=\frac{{\left|\bar{y_{i}}-\overset{}{y_{i}}\right|}_{\max}}{Y_{FS}}\times X_{FS}\times 100\%$, where $\bar{y_{i}}$ is the average of the output of the samples in the entire measuring range.

## 3. Results and Discussion

To measure the change in the accuracy of the pressure sensor before and after processing, a pressure sensor test system was established, as shown in Figure 3, which was composed of a pressure measurement and control unit (including the high-precision standard pressure sensor, the pump, the valve, the computer), an electrical parameters measurement unit (including the power supply and the multimeter), and a sealed pressure chamber. The system is able to realize the measurement of the main indexes, such as the non-linearity error and sensitivity of the pressure sensor.

The nonlinear error is one of the most important indexes which can be used to measure the accuracy of pressure sensors. Measurement accuracy is generally defined as the consistency between the measured value and the real value of the measured object [27]. For the silicon piezoresistive linear pressure sensor studied in this paper, its true value should be an ideal straight line of an input–output relationship, and the non-linear error of the characteristic curve of the piezoresistive linear pressure sensor is expressed by the maximum deviation between the characteristic curve and its fitting straight line, so the non-linear error is the main factor affecting the accuracy of silicon piezoresistive linear pressure sensors. Therefore, the non-linear error was used to represent the accuracy of the silicon piezoresistive linear pressure sensor in this paper [28,29,30,31].

After testing and data processing, the sensor-characteristic curve of the pressure sensor before the accuracy-improving, post-processing method is shown in Figure 4a. The maximum value of the device’s standard deviation is 0.058 mmHg. After the accuracy-improving, post-processing method, the sensor-characteristic curve of the pressure sensor is shown in Figure 4b. Moreover, the value of the device’s standard deviation went down to 0.016 mmHg, which was a reduction by 72.41%. Besides, the deposition of the film caused a small increase in the sensor’s offset. It seems that the dynamic fluctuations of temperature and humidity, and the specific stress of the film deposition on the device surface resulted in a small increase of the sensor offset. The sensor offset could be solved by compensating.

Using the above test system, we obtained the changes in accuracy and sensitivity of the pressure sensors before and after the processing cladded with the micro-coating. As shown in Figure 5a, experiments on seven pressure sensors were conducted, in which the micron-thin parylene polymer was utilized as the surface-deposited coating layer to demonstrate the improvement in accuracy. Furthermore, as shown in Figure 5b, the error decreased, on average, by 62.54%, so the accuracy was improved. The accuracy of all pressure sensors before and after processing treatment and their change rates are shown in Table 1. In order to reduce the system error in the experiment, the pressure change in the pressure chamber of the test system was controlled within 1 Pa, corresponding to 0.0075 mmHg, which is enough to ensure the reliability of the experimental results.

In fact, the experiments and tests were done with some interfering quantities. Although the experiments were carried out in a relatively ideal environment, it was not an environment with absolutely constant temperature and humidity. Conversely, the temperature fluctuated dynamically within the range of ±0.1 °C, and the relative humidity fluctuated dynamically within the range of 40% to 80%. If the sensors were not processed the proposed parylene-C coating treatment in this paper, these dynamic fluctuations should result in a random error evaluated to be about ±0.23 mmHg to the sensors. However, after the proposed parylene-C coating treatment in this paper, the accuracy of the sensors obviously improved under the same interference environment, as shown in Figure 5a and Table 1. Therefore, it demonstrates that the parylene-C coating deposited film can protect the sensors from the environment interfering quantities such as temperature and humidity.

Besides that, the sensitivity was almost unchanged. The calculation method of sensitivity was as follows:

The formula for calculating sensitivity:$$k=\frac{m\sum_{i=1}^{m}X_{i}\bar{Y_{i}}-\sum_{i=1}^{m}X_{i}\sum_{i=1}^{m}\bar{Y_{i}}}{m\sum_{i=1}^{m}X_{i}^{2}-{\left(\sum_{i=1}^{m}X_{i}\right)}^{2}}$$

In the above formula: $X_{i}$ is the input of pressure value of the test point *i* (*i* = 1, 2, 3,…, *m*); $\bar{Y_{i}}$ is the average of the output of positive and negative travel of the test point *i*; and *m* is the number of test points.

The formula for calculating the change rate of sensitivity was:$$\Delta k=\frac{k_{2}-k_{1}}{k_{1}}$$
where *k*_1_ is the device sensitivity before processing, and *k*_2_ is the device sensitivity after processing. The experimental data and data-processing results are shown in Table 2. The sensitivity of the processed device was reduced, on average, by 0.90%. Therefore, the processing had little effect on the sensitivity of the device.

Besides, the effect of the SO8 packaging on the parylene-coating process is also considered, so we did a comparative experiment of parylene-coating sensors with the SO8 packaging and the removal of the SO8 packaging. After testing, we found that the removal of SO8 packaging had no significant impact on the performance of the devices by the process treatment of this paper. The reason for this is that the SO8 packaging of SMI SM5240 contains a millimeter through-hole to sense the pressure signal. During the deposition of the parylene coating, the gaseous parylene will enter the package through this hole and deposit on the surface of the sensitive area of the pressure-sensitive chip. Therefore, after the coating process, the parylene will evenly clade the whole surface of the pressure-sensitive chip of the SMI SM5240 sensor with the SO8 packaging.

Results showed that the accuracy of the pressure sensor was improved through the proposed parylene micro-coating post-treatment, because the micro-coating enhanced the accuracy of the pressure sensor. Firstly, the micro-coating of parylene has good barrier protection ability, which isolates the influence of media, such as water molecules and oxygen, that affect the performance of pressure sensors in external environments. Secondly, the parylene coating with a low calorific conduction coefficient reduces the thermal effect of the temperature on the pressure sensor. Finally, the lightweight micro-coating has little effect on the flexibility of the movable structure of the pressure sensor and the output of the sensor signal. Therefore, the post-processing method of cladding micro-coating can reduce the errors of the sensor, resulting in a more accurate output.

## 4. Conclusions

In summary, a simple, efficient, and low-cost method was proposed to improve the accuracy of the pressure sensor while maintaining the sensitivity of the device. The mechanism of improving accuracy by micro-coating has been revealed: a micro-coating is cladded on the surface of the sensor, which effectively isolates the adverse effect of the external environment. Moreover, the ultra-thin coating layer has negligible effects on the sensitivity of the sensor. Results showed that the accuracy of the pressure sensor was increased by 62.54% after applying the proposed method, while the sensitivity was almost unchanged. Besides that, this post-processing method based on micro-coating of parylene offers the following advantages: (1) With the coating as a protective barrier, the device can be well-isolated from the external environment’s media, such as the surrounding gas, acid, alkali, and water. (2) The coating is lightweight, which can hardly affect the size of the device. (3) This post-processing method is simple and independent, which exerts negligible effects on the process operations before and after it. (4) This low-cost post-processing method can be produced at a large scale for improving the accuracy of the device. (5) This method has excellent performance in waterproofing, corrosion resistance, and biocompatibility. Therefore, this simple, efficient, and low-cost method for improving the accuracy of pressure sensors has desirable application prospects in industrial production [32,33], biochemical detection, medical diagnosis [34], marine exploration [35], environmental protection, experimental testing, and other industrial automation. This method should also be useful for improving the accuracy of other kinds of sensors.

## Figures and Tables

**Figure 1 sensors-19-04601-f001:**
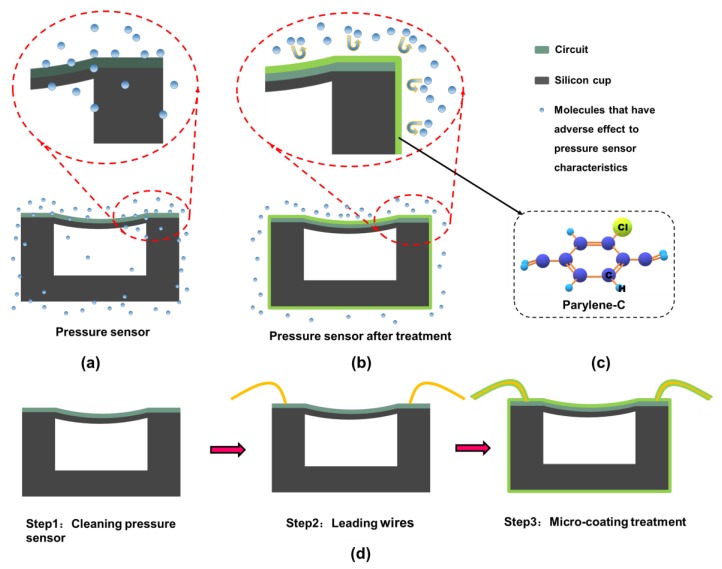
(**a**) A pressure sensor influenced adversely by environmental molecules, affecting the accuracy of the device. (**b**) A pressure sensor with the micro-coating to resist the adverse influence of molecules affecting the accuracy of the device. (**c**) The chemical molecular structure of the micro-coating: parylene-C. (**d**) Three steps of the sensor treatment in this experience.

**Figure 2 sensors-19-04601-f002:**
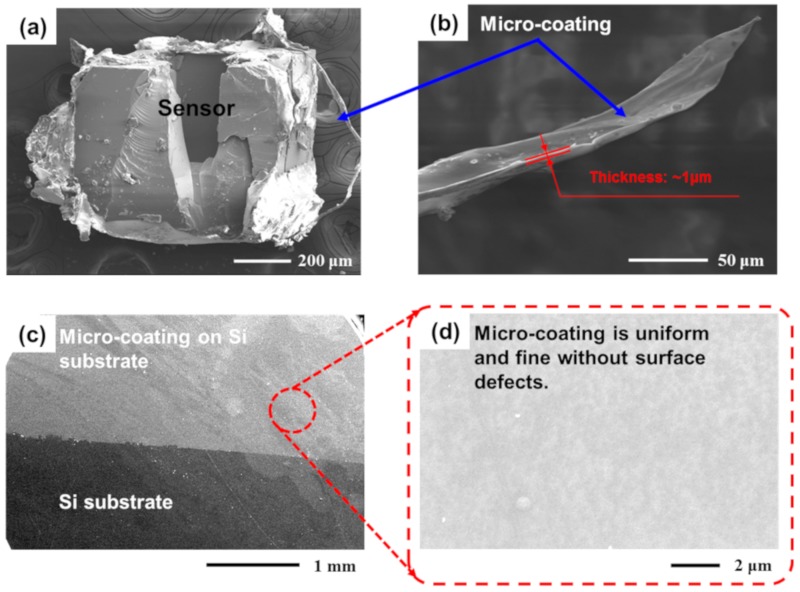
(**a**) A scanning electron microscopy (SEM, Sirion 200, 10 kV) image on the profile of a cleaved sensor coated with the micro-layer. It indicates that the micro-coating is very lightweight. (**b**) SEM image on the side view of the micro-coating. (**c**,**d**) Top-view SEM images of the micro-coating on silicon substrates showing that the coating is uniform and fine without surface defects.

**Figure 3 sensors-19-04601-f003:**
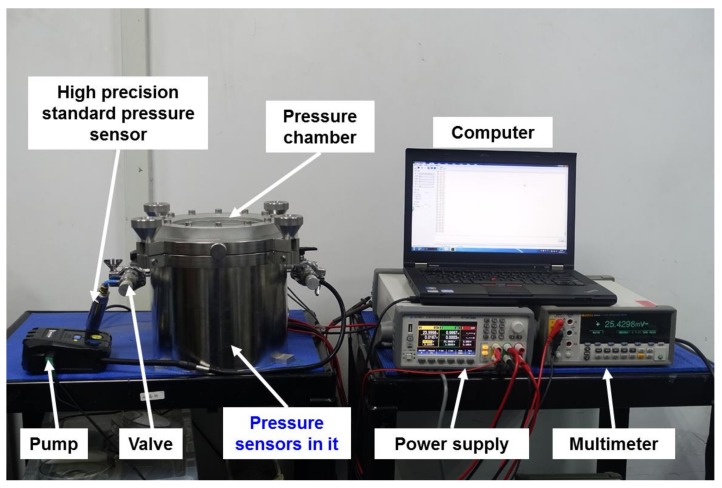
Photograph of the pressure sensor test system.

**Figure 4 sensors-19-04601-f004:**
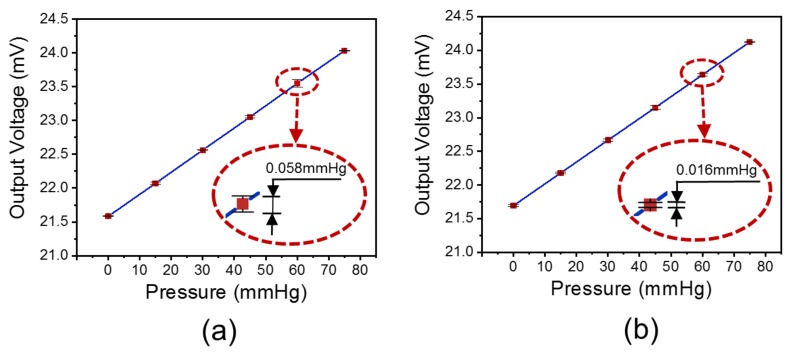
(**a**) The voltage-pressure characteristic curve of the pressure sensor before the accuracy-improving, post-processing method. (**b**) The voltage-pressure characteristic curve of the pressure sensor after the accuracy-improving, post-processing method.

**Figure 5 sensors-19-04601-f005:**
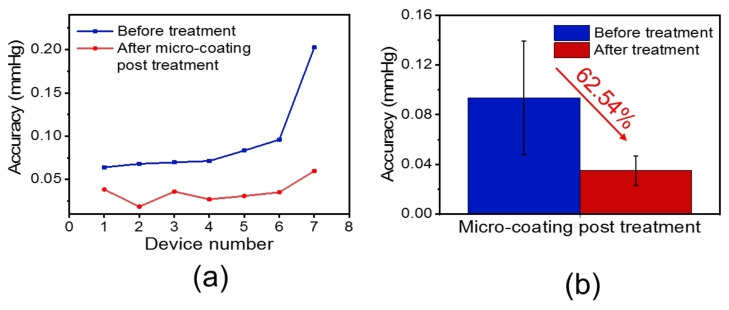
(**a**) Accuracy of the devices before and after applying the micro-coating cladding post-treatment. (**b**) After employing the proposed micro-coating post-treatment method, the accuracy of the whole devices decreased by approximately 62.54%.

**Table 1 sensors-19-04601-t001:** Accuracy of pressure sensors before and after the processing treatment and their change rates.

Device Number	Accuracy Before Processing (mmHg)	Accuracy After Processing (mmHg)	Change Rate of Accuracy
1	0.063948	0.038288	−0.401264
2	0.067927	0.018481	−0.727925
3	0.069682	0.035841	−0.485652
4	0.071272	0.026932	−0.622122
5	0.083141	0.030785	−0.629732
6	0.095768	0.035051	−0.634001
7	0.202403	0.059648	−0.705302
Average	0.093449	0.035004	−0.625424

**Table 2 sensors-19-04601-t002:** Sensitivities of pressure sensors before and after processing treatment and their change rates.

Device Number	Sensitivity Before Processing (mV/mmHg)	Sensitivity After Processing (mV/mmHg)	Change Rate of Sensitivity △*k*_i_
1	0.267143333	0.264535952	−0.009760232
2	0.224944048	0.221267619	−0.016343749
3	0.227907143	0.231348333	0.015099088
4	0.251315952	0.249302381	−0.00801211
5	0.244220714	0.241308571	−0.011924226
6	0.236820714	0.231635952	−0.021893195
7	0.245490714	0.242974048	−0.010251573
△*k*			−0.009012285

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
