# Peer review of "A Simple, Low-Cost Micro-Coating Method for Accuracy Improvement and Its Application in Pressure Sensors"

_sensors, 2019, doi:10.3390/s19204601_

Round 1

Reviewer 1 Report

Yao et al. propose a low-cost method to improve the performances of pressure sensors. The strategy is based on the deposition of a thin parylene film on the surface of the sensing structure. The work is interesting and the manuscript is well organized. The results confirm that some properties of the sensor have been improved.  Nevertheless, a more critical approach should be used in the result discussion and some technical aspects should be clarified.

In particular, the authors claim that the deposited film protects the sensors from many environment impurities and other interfering quantities such as temperature and humidity. However no experimental results have been reported about this aspect and tests have been performed in an ideal circumstances, e.g. inside a clean pressure chamber. In my opinion, the authors should investigate the protection from some interfering quantities, such as temperature and/or humidity, in order to demonstrate the usefulness of the parylene film.

Furthermore, the authors use the term accuracy in a non-standard way. The measurement accuracy is usually defined as the closeness of agreement between the measured quantity value and the true quantity value of the measurand. The authors define the accuracy as the non-linear error but this is a property regarding the linearity of the sensor not its accuracy. The author should clarify better this aspect using for the quantities the definitions usually accepted in the scientific community.

The author should clarify the method used to calculate the sensitivity of the sensor and they should specify what experimental data they used to demonstrate that the sensitivity decrease is about 0.9%.

The deposition of the film seems to cause also a small increase of the sensor offset, e.g. the output voltage for zero pressure (see Figure 4). Is this effect present in all the tested sensors? Which are the causes? The authors should discuss about this issue.

Author Response

Thank you for your detailed comments and the time you spent. Please see the attachment.

Reviewer 2 Report

In the page 6, line 158, it is written that the accuracy was increased by 62.54% on average due to treatment. However, from the Figure 5 (b) looks like accuracy is decreased after treatment. Please clarify this.

Author Response

Thank you for your constructive comments and the time you spent. Please see the attachment.

Reviewer 3 Report

This article details using a parylene coating to improve the accuracy of piezoresistive pressure sensors. These coating work to protect the sensor from adverse environmental effects due to its inherent chemical inertness as well as a very low temperature conductivity. It is felt that there should be some significant revisions made to this paper as some of the conclusions do not necessarily seem to be supported by the data. A more thorough analysis of the data should be conducted to support the improvements stated by the authors. As for quality of presentation, some minor editing should be performed to improve clarity, but it is not distracting to the reader. The specific questions and concerns are listed below.

Page 2, Lines 75-77: the sentence beginning "to meet the above three requirements...." should be reworded as the word selected is used too many times. Page 3, Materials and Methods Section: the authors are citing the use of SMI SM5240 sensors. However, they do not state the specific range of the sensors. From the SMI website, the nearest product is the SMI 5420E, a series of absolute pressure sensors. These sensors are sold in ranges from 30-100 psi ranges. Furthermore, they are sold in SO8 packaging. Are the authors actually using this sensor, or are they simply using the SM5108E MEMS piezoresisitve die (which is the actual pressure transducer in the SM5240E). This is an important distinction, as some of the processing steps do not make sense if the full SM5240E are being used. For example, is the SM5240 package is being used, do the researchers have to cut away any of the packaging before depositing the parylene? Furthermore, if the SM5240 is being used, why is there a need for adding wires to interrogate the response, as the SO8 packaging includes ways to measure and power the sensor? Page 4, Section 2.3: this can also be clarified if the authors are using the SMI 5240 or the SMI 5108 membrane. The authors state that the characterization of the pressure sensor was done by making measurements at 6 pressures that were selected within the full range of the sensor, with the pressure measurements ranging from 0 to 75 mm Hg. However, the smallest range of the SM5240 is 0 to 30 psi, corresponding to 0 to 1551 mm Hg. Pages 4-5, Results and Discussion: Figure 4 and the conclusions drawn from this are problematic. This figure is used to state that the addition of the parylene cladding decreases the maximum value of the devices standard deviation from 0.058 mm Hg to 0.016 mm Hg. However, this occurs at only one point of the six that are measured. Furthermore, which the figure itself is small, it can be argued that at 2 of the measured pressures, the addition of the cladding made the standard deviation larger. This also brings into question the results from Figure 5 as there is not a clear explanation of how these accuracy values were derived. Was it simply looking at changes in standard deviation? If so, then it might be prudent to report on any improvements in accuracy at each pressure as it seems that the standard deviation is not consistent over the pressures. Furthermore, are there any systematic causes for the increased standard deviation at some points? Could the pressure of the test chamber be varying a small amount? There may be other explanations for the observed improvements from the cladding.

Author Response

(The authors gave the same response as above.)

Round 2

Reviewer 1 Report

Yao et al. addressed the issues proposed by the referee in the author response file but they have only partially modified the manuscript. In particular, they have modified the text only to clarify the issue regarding the test conditions. In my opinion the authors should revised the manuscript including the discussion and the explanations that they have given in the author response file.

In particular, they should revise the text in order to address the following issue:

1) why did they used the non-linear error to characterize the sensor accuracy? (By the way, the definition of accuracy suggested to the authors was taken from the “International vocabulary of metrology – Basic and general concepts and associated terms (VIM), 3rd edition edited by the Joint Committee for Guides in Metrology (JCGM) and not by a simple textbook).

2) The methods used to calculate the accuracy and the sensitivity should be described in the text.

3) The discussion about the sensor offset should be added to the text and not be postponed to a future paper.

Author Response

(The authors gave the same response as above.)

Reviewer 3 Report

While this is recommended for publication after another major revision, this is simply to include several of the remarks the authors' made in response to the original review report. While the authors' satisfied the report, they did not modify the manuscript to include their answers.

Point 1: Thank you. That sentence makes more sense now.

Point 2: Thank you for the clarification, especially about the use of the exterior packaging as well as the inclusion of the additional wires. While it is recognized that this would add some length to the manuscript, it is felt that at least a summary of the authors' response to this point should be included to minimize confusion from other readers. Specifically, the statements that the response improvement was seen regardless of the external packaging.

Point 3: Thank you.

Point 4: This is an excellent summary of how the authors' came to their conclusions about the improvement to accuracy using the parylene coating. However, as with Point 2 above, it is felt that this should be included in the final manuscript, including the table that was provided. While it is recognized that this will add even more length to the manuscript, it is critical information to the manuscript.

Author Response

Thank you for your professional suggestions and the time you spent. Please see the attachment.

Round 3

Reviewer 1 Report

The authors have adequately dealt with all previous remarks and they have modified the manuscript according to the reviewer comments.

Reviewer 3 Report

Thank you for making the suggested additions. It is my recommendation that this paper be published.